# Three previously unrecognised classes of biosynthetic enzymes revealed during the production of xenovulene A

Raissa Schor[1,2], Carsten Schotte[1,2], Daniel Wibberg[3], Jörn Kalinowski[3] & Russell J. Cox[1,2]

Xenovulene A is a complex fungal meroterpenoid, produced by the organism hitherto known as *Acremonium strictum* IMI 501407, for which limited biosynthetic evidence exists. Here, we generate a draft genome and show that the producing organism is previously unknown and should be renamed as *Sarocladium schorii*. A biosynthetic gene cluster is discovered which bears resemblance to those involved in the biosynthesis of fungal tropolones, with additional genes of unknown function. Heterologous reconstruction of the entire pathway in *Aspergillus oryzae* allows the chemical steps of biosynthesis to be dissected. The pathway shows very limited similarity to the biosynthesis of other fungal meroterpenoids. The pathway features: the initial formation of tropolone intermediates; the likely involvement of a hetero Diels–Alder enzyme; a terpene cyclase with no significant sequence homology to any known terpene cyclase and two enzymes catalysing oxidative-ring contractions.

[1] Institute for Organic Chemistry, Leibniz University of Hannover, Schneiderberg 1B, 30167 Hannover, Germany. [2] BMWZ, Leibniz University of Hannover, Schneiderberg 38, 30167 Hannover, Germany. [3] Center for Biotechnology - CeBiTec, Universitätsstraße 27, 33615 Bielefeld, Germany. Correspondence and requests for materials should be addressed to R.J.C. (email: russell.cox@oci.uni-hannover.de)

Xenovulene A **1** (Fig. 1a) is an unusual meroterpenoid produced by the fungus *Acremonium strictum* IMI 501407 (also known as *Sarocladium strictum*)[1]. It has potent (40 nM) inhibitory effects vs. the human γ-aminobutyrate A (GABA_A) benzodiazepine receptor and potential use as an anti-depressant with reduced addictive properties[2]. Previous biosynthetic studies using stable isotopes, reported by Simpson and coworkers, showed that **1** is derived from humulene (α-caryophyllene) **2** bonded to a rearranged polyketide-derived moiety[3]. It was hypothesised that a ring-expansion/ring-contraction mechanism could transform a methyl-orsellinic acid or aldehyde into the cyclopentenone of **1** via proposed tropolone intermediates (Fig. 1b). Later, we showed that fungi produce tropolones such as stipitatic acid **3** via oxidative-ring expansion of methylorcinaldehyde **4** using a two-step process in which

oxidative dearomatisation by an FAD-dependent enzyme (TropB) is followed by methyl-oxidation catalysed by a non-haem iron oxygenase (TropC) that catalyses the ring expansion to give **5** (Fig. 1b)[4]. Oxidative-ring contractions must occur in other fungal biosynthetic pathways, for example, during the biosynthesis of terrein **6** in *Aspergillus terreus* (Fig. 1c)[5], but no evidence linking genes, proteins and chemical steps is currently available[6].

In an early attempt to gain molecular information on xenovulene biosynthesis, we previously obtained a short genomic fragment of *A. strictum* IMI 501407 containing a partial biosynthetic gene cluster (BGC) from which one gene (*aspks1*) was shown to encode an iterative non-reducing polyketide synthase (nr-PKS) which produces 3-methylorcinaldehyde **4** when expressed in *Aspergillus oryzae*[7]. However, we were unable to extend the gene cluster, or perform effective knockouts in wild-type (WT) *A. strictum* IMI 501407, and further progress was halted. Here, we report the results of genome sequencing of *A. strictum* IMI 501407, the discovery of xenovulene A **1** BGC, heterologous expression experiments to probe the biosynthetic steps and the discovery of three previously unreported types of biosynthetic enzymes, including a putative hetero-Diels Alderase, two enzymes that catalyse oxidative-ring contractions and a class of terpene cyclase unrelated to other known terpene cyclases.

## Results

**Analysis of WT *A. strictum* IMI 501407.** *A. strictum* IMI 501407 was obtained from the Centre for Agriculture and Bioscience International (CABI), and under literature fermentation conditions, the production of a compound corresponding to **1** was observed by liquid chromatography mass spectrometry (LCMS). Purification and ¹H nuclear magnetic resonance (NMR) characterisation proved this to be xenovulene A as expected. A series of related compounds **7**–**13** was also observed (Fig. 2a, b, Supplementary Figs. 1–31, Supplementary Tables 1–5). Purification and full NMR analysis of **7** (1 mg), and its isomer **8** (<0.5 mg) showed these to be phenolic meroterpenoids. Full NMR assignment of **9** (113 mg) and **10** (19 mg) revealed them as previously reported, but uncharacterised, tropolone meroterpenoids.

Two unreported tropolone-containing metabolites, **11** and **12**, were also isolated (9 mg and 5 mg, respectively) and fully

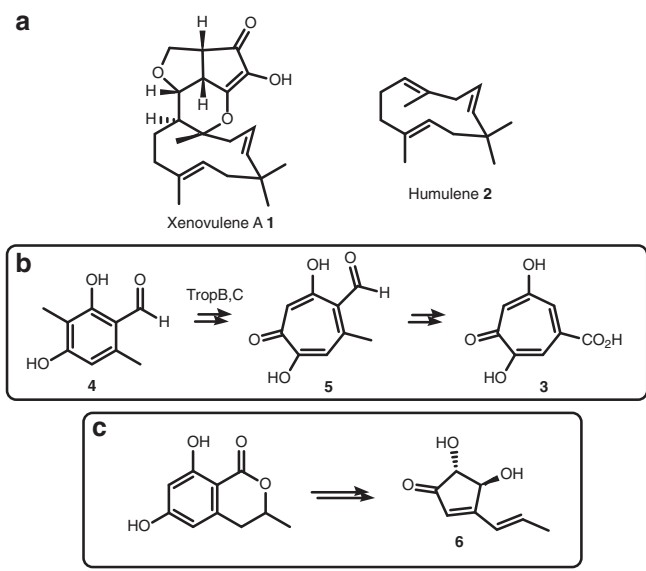

**Fig. 1** Xenovulene and related compounds. **a** Structures of xenovulene A **1** and humulene **2**; **b** fungal route to tropolones from methylorcinaldehyde **4** involving oxidative-ring expansion and **c** fungal route to terrein **6** involving ring contraction

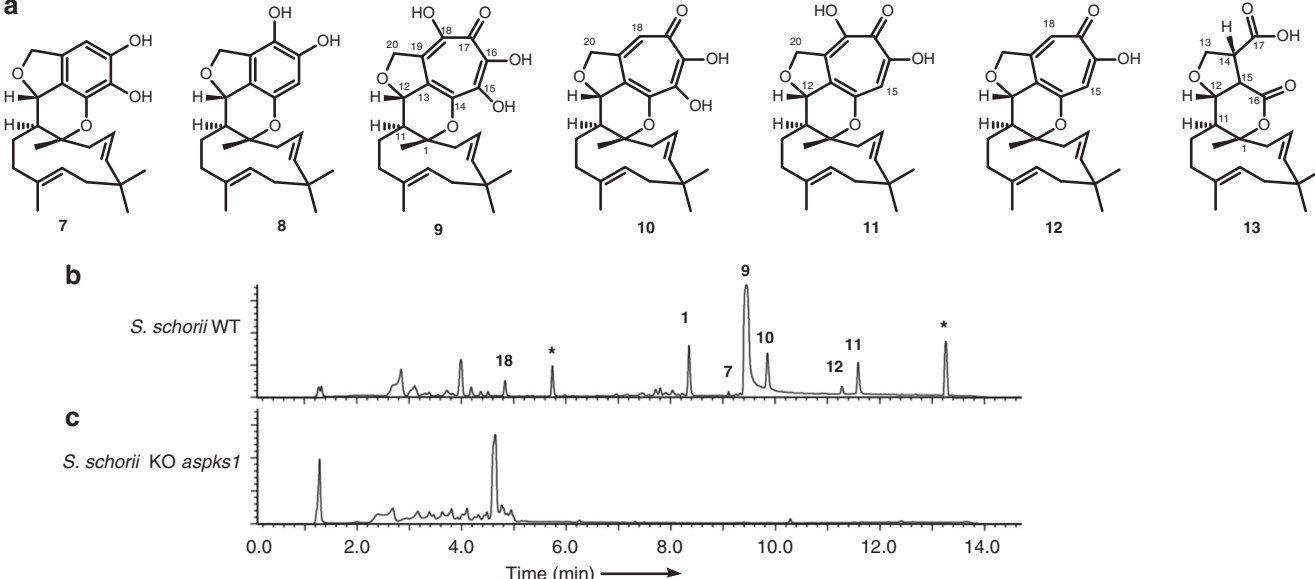

**Fig. 2** Isolation of wild-type compounds. **a** Compounds isolated from *S. schorii*; **b** HPLC (diode array detector) trace of organic extracts from WT *S. schorii*, * = unrelated compound and **c** HPLC (diode array detector) trace of organic extracts from *S. schorii* Δ*aspks1*

characterised. Comparison of high-resolution mass spectrometry (HRMS) data ([M]H$^+$ calculated C$_{24}$H$_{31}$O$_5$ 399.2171, found **10**: 399.2168 and **11**: 399.2166) and UV absorption data for **10** and **11** suggested these to be structural isomers. By comparing highly similar $^1$H and $^{13}$C chemical shifts, small differences in the resonances of the aromatic protons (7.12 at $\delta_C$ 108.9 in **10** and 7.16 at $\delta_C$ 112.3 in **11**) were detected. Further analysis of heteronuclear multiple-bond correlation (HMBC) data of **10** showed a $^3J_{CH}$ correlation of the aromatic proton to C-20 and thus the 7.12-ppm proton was proposed to be at position 18. HMBC data of compound **11** showed no such correlation between the aromatic proton and C-20, but instead a $^4J_{CH}$ interaction between the aromatic proton and C-12 ($\delta_C$ 84.3 ppm) was detected which indicates its attachment at position 15 (Supplementary Figs. 12–21 and Supplementary Table 3).

HRMS of the second unknown compound **12** suggested a chemical formula of C$_{24}$H$_{29}$O$_4$ ([M–H]$^-$ calculated 381.2066, found 381.2062), which corresponds to a less-hydroxylated tropolone. Analysis of $^1$H and $^{13}$C NMR data confirmed its similarity to **9–11** and showed two aromatic C–H fragments (6.96 at $\delta_C$ 110.1, 6.91 at $\delta_C$ 113.1). HMBC correlations of **12** showed a $^3J_{CH}$ coupling of the 6.91-ppm proton with C-20 ($\delta_C$ 75.2 ppm) and a $^4J_{CH}$ interaction between the 6.96 proton and C-12 ($\delta_C$ 84.2 ppm, Supplementary Figs. 22–26 and Supplementary Table 4). These observations indicate that the 6.96- and 6.91-ppm protons are at positions 18 and 15, respectively.

$^1$H NMR of **13** (C$_{21}$H$_{30}$O$_5$, [M]H$^+$ calculated 363.2171, found 363.2179) clearly showed the presence of the humulene skeleton substituted in the usual way, and chemical shift analysis confirmed the presence of the tetrahydrofuran ring and loss of the cyclopentenone ring in comparison to **1**. HMBC correlations, particularly from H-12 and H-15 to lactone carbonyl C-16, and from H-13ab to carboxylic acid C-17 confirmed the structure.

### Genome and transcriptome analysis of *A. strictum* IMI 501407.

Genomic DNA was prepared from the organism known as *A. strictum* IMI 501407 by extraction from mycelia using phenol/chloroform, then purified by caesium chloride density-gradient centrifugation and sequenced using Illumina paired-end technology. Sequence assembly was achieved using gsAssembler 2.8

(Roche Diagnostics, Mannheim, Germany) to afford a draft genome of ~33.8 Mb with scaffold N50 of 1.3 Mb (Supplementary Table 10). Average nucleotide identity (ANI) comparison[8] of the draft genome sequence with previously obtained genomes of *Acremonium/Sarocladium* species (*S. strictum*, *Sarocladium kiliense*)[9] showed that the organism is likely to be a previously unreported species of the genus *Sarocladium*, because ANI results were below 97%[10]. All comparison with other species gives results of around 80% ANI. We propose the name *Sarocladium schorii* for this organism.

AntiSMASH analysis[11] suggested the presence of at least 39 secondary metabolite BGC (Supplementary Fig. 46). Basic Local Alignment Search Tool (BLAST) searching using the previously identified *aspks1* gene rapidly identified a 1.6-Mb scaffold as containing a 49-kb BGC potentially involved in tropolone biosynthesis (Table 1). Automatic gene prediction by Augustus 3.0[12] and annotation within the GenDBE platform[13] revealed the presence of numerous genes with potential roles in secondary metabolism (Supplementary Fig. 60, Supplementary Table 11). Previous work has shown that the key steps of fungal tropolone biosynthesis are catalysed by a non-reducing PKS with a reductive release domain (TropA); an FAD-dependent salicylate hydroxylase (TropB); a non-haem iron dioxygenase (TropC) and a cytochrome P450 monooxygenase (TropD)[4]. Homologues of all four proteins are encoded by this BGC (*aspks1*, *asL1*, *asL3* and *asR2*, respectively), and no other annotated BGC from the *A. strictum* genome encodes a similar manifestation of enzymes (Table 1, Fig. 3). The *aspks1* BGC also encodes many other potential tailoring proteins, transporters and transcription factors, although no significant levels of homology to other known secondary metabolism tailoring proteins were observed among them, and potential cluster boundaries were obscure.

In order to link the *aspks1* cluster to the biosynthesis of **1**, targeted knockout (KO) experiments were attempted. A transformation protocol, involving the generation of *S. schorii* protoplasts and insertion of a hygromycin-resistant cassette consisting of the *A. nidulans gpdA* promoter ($P_{gpdA}$) fused to the *E. coli hph* gene[14] was developed. The bipartite knockout method of Nielsen and co-workers was used to target *aspks1*[15]. Forty-three transformants were generated and of these, only one was shown to be a true *aspks1* knockout by polymerase chain reaction

---

**Table 1 Annotation of genomic region surrounding *aspks1***

| ORF | bp/AA | Proposed function | ORF | bp/AA | Proposed function |
|---|---|---|---|---|---|
| *asL7* | 4350/1449 | ABC transporter | *AsPKS1* | 8190/2729 | 3-Methylorcinaldehyde synthase (MOS) |
| *asL6* | 1272/423 | NAD/FAD-dependent oxidoreductase | *asR1* | 1308/435 | MFS |
| *asL5* | 750/249 | SDR—short-chain dehydrogenase | *asR2* | 1551/516 | Cytochrome P450 |
| *asL4* | 1293/430 | NAD/FAD-dependent oxidoreductase | *asr3* | 2565/854 | Transcriptional regulator |
| *asL3* | 1023/340 | Non-haem iron II oxygenase | *asR4* | 1899/632 | Unknown |
| *asL2* | 348/115 | Ferredoxin like[a] | *asR5* | 1206/401 | Unknown |
| *asL1* | 1443/480 | FAD-dependent salicylate hydroxylase | *asR6* | 1293/430 | Unknown |
| | | | *asR7* | 2469/822 | Unknown |

[a]Phyre$^2$ analysis[41]

---

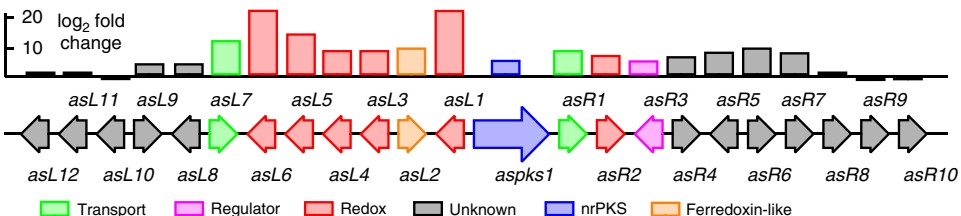

**Fig. 3** Transcription analysis of the xenovulene BGC. Approximate values, see Supplementary Table 14 for actual values

(PCR). LCMS analysis of this transformant showed the loss of production of xenovulene A **1** and the related compounds **7**–**13** (Fig. 2c). Complementation of the KO by insertion of *aspks1* driven by the *A. oryzae amyB* promoter ($P_{amyB}$) restored production of **1**. Attempts to knockout other nearby genes were universally unsuccessful. We[16] and others[17] have shown that gene silencing by RNA interference can be effective in filamentous fungi, but again, in the case of *S. schorii*, satisfactory results could not be achieved. Likewise, extensive efforts were made to utilise CRISPR/Cas9 methods which have been shown to be effective in several fungal species[18], but without satisfactory outcome.

In an effort to obtain further information about the xenovulene BGC, we made a transcriptomic comparison of expression levels of genes surrounding *aspks1* under producing and non-producing conditions (Supplementary Table 13). Total RNA was collected from fermentations of *S. schorii* grown in ASPM (production) and DPY (non-production) media and sequenced by standard Illumina methods. Tophat2 was used to map quality-filtered reads onto the *S. schorii* draft genome[19]. Differential gene expression was then analysed with ReadXplorer 2.0[20] including the DESeq package[21]. The results (Fig. 3) clearly show that genes from *asL7* to *asR7*, inclusive, are significantly upregulated under producing conditions, while genes outwith this area show no significant change in transcription level. Furthermore, close inspection of the transcriptome data allowed all intron positions and all transcriptional start and stop positions to be unambiguously determined for the genes of interest.

**Heterologous reconstruction of tropolone biosynthesis**. Heterologous expression experiments were conducted to confirm the function of the proposed early genes. Expression of *aspks1* alone in the fungal host *A. oryzae* NSAR1[22] results in the synthesis of polyketides **4** and **14** (Table 2 exp. 1, Supplementary Fig. 50) in agreement with previous results[23]. Addition of *asL1* (encoding a salicylate hydroxylase, Table 2 exp. 2) leads to the oxidative dearomatisation of **4**, giving **15** and a tautomer **15a**. Addition of the non-haem iron oxygenase encoded by *asL3* (Table 2 exp. 3) gives the tropolones **5** and **16**; and addition of the cytochrome P450 monooxygenase (Table 2 exp. 4, Fig. 4) encoded by *asR2* allows the synthesis of **18**–**21** in varying amounts presumably via lactol **17** (Table 2, entries 1–4, Supplementary Figs. 51–53)[24].

**Full pathway expression**. In the absence of useful knockout data, we constructed a series of fungal expression plasmids using the modular system described by Lazarus and co-workers[25]. These included all genes from the complete putative cluster, excluding those encoding transporters (*asL7* and *asR1*) and the transcription factor (*asR3*) and the unknown protein encoded by *asR7*. The plasmids were transformed into *A. oryzae* NSAR1. This host organism cannot produce xenovulene **1**, but growth of the transformants under inducing conditions led to the clear production of **1** by LCMS analysis (exp. 5, Fig. 4), in addition to the usual manifest of intermediates and shunts. Purification of **1** from this experiment and full analysis by NMR (Supplementary Figs. 37–45) unambiguously confirmed this observation. A further meroterpenoid, **22**, was also isolated and fully characterised from this strain ($C_{23}H_{31}O_6$ [M–H]$^-$, calculated 403.2121, found 403.2129, Supplementary Table 7 and Supplementary Figs. 32–36). $^1$H NMR showed the intact humulene with the usual substitution pattern and an intact tetrahydrofuran ring. HMBC correlations, particularly from H-13ab to carbonyl C-15 located the hydroxyacetyl unit as attached to C-14, while chemical shift correlations were used to place carboxylic acid C-19 as attached to C-18.

We then deployed a knockout by expression (KOe) strategy in which individual genes, and groups of genes were omitted from the sets of expressed genes. Results of this analysis showed that *asL5*, *asR4* and *asL2* (Table 2, entries 6–8, respectively) are not required for the biosynthesis of **1**. In order to show that these genes cannot complement one another, the minimal gene set lacking these three genes was constructed and expressed in *A. oryzae* and this also showed successful production of **1** (exp. 9, Fig. 4). The production of **1** was also accompanied by **22**, which is likely to be a shunt or by-product of the pathway in *A. oryzae*. Thus, the minimal set of genes responsible for the biosynthesis of **1** consists of *aspks1*; *asL1* encoding an FAD-dependent hydroxylase; *asL3* encoding a non-haem iron dioxygenase; *asR2* encoding a cytochrome P450 monooxygenase; *asR5* and *asR6* encoding proteins of unknown function and *asL6* and *asL4* encoding putative NAD/FAD-dependent oxidoreductases.

**Determination of chemical steps**. Further KOe experiments among the minimal gene set showed that the unknown genes *asR6* and *asR5* encode proteins that are involved in production and/or linkage of the humulene and the polyketide moiety (Table 2, exp. 10 and 11). Lack of either of these genes results in *A. oryzae* strains incapable of meroterpenoid production,

**Table 2 Combinations of genes from the xenovulene BGC used for heterologous expression experiments in *A. oryzae***

| Exp. | Gene annotation | aspks1 PKS | asL1 FAD | asL3 NHI | asR2 P450 | asL2 ? | asL4 FAD | asL5 SDR | asL6 FAD | asR4 ? | asR5 hDA | asR6 Hum | Metabolites |
|---|---|---|---|---|---|---|---|---|---|---|---|---|---|
| 1 | | ✓ | – | – | – | – | – | – | – | – | – | – | 4, 14 |
| 2 | | ✓ | ✓ | – | – | – | – | – | – | – | – | – | 4, 14, 15 |
| 3 | | ✓ | ✓ | ✓ | – | – | – | – | – | – | – | – | 4, 5, 14–16 |
| 4 | | ✓ | ✓ | ✓ | ✓ | – | – | – | – | – | – | – | 4, 14–16,18,19, 21 |
| 5 | *Full* | ✓ | ✓ | ✓ | ✓ | ✓ | ✓ | ✓ | ✓ | ✓ | ✓ | ✓ | **1**, 4, 14–16,18-**22** |
| 6 | *FullΔL5* | ✓ | ✓ | ✓ | ✓ | ✓ | ✓ | – | ✓ | ✓ | ✓ | ✓ | **1**, 4, 14–16,18-**22** |
| 7 | *FullΔR4* | ✓ | ✓ | ✓ | ✓ | ✓ | ✓ | ✓ | ✓ | – | ✓ | ✓ | **1**, 4, 14–16,18-**22** |
| 8 | *FullΔL2* | ✓ | ✓ | ✓ | ✓ | – | ✓ | ✓ | ✓ | ✓ | ✓ | ✓ | **1**, 4, 14–16,18-**22** |
| 9 | *Minimal* | ✓ | ✓ | ✓ | ✓ | – | ✓ | – | ✓ | – | ✓ | ✓ | **1**, 4, 14–16,18-21, **22** |
| 10 | *FullΔR5* | ✓ | ✓ | ✓ | ✓ | ✓ | ✓ | ✓ | ✓ | ✓ | – | ✓ | 4, 14–16,18-21 |
| 11 | *FullΔR6* | ✓ | ✓ | ✓ | ✓ | ✓ | ✓ | ✓ | ✓ | ✓ | ✓ | – | 4, 14–16,18-21 |
| 12 | *FullΔL6* | ✓ | ✓ | ✓ | ✓ | ✓ | ✓ | ✓ | – | ✓ | ✓ | ✓ | **1**$^a$, 4, **7**$^a$, 14–16,18-21 |
| 13 | *FullΔL4* | ✓ | ✓ | ✓ | ✓ | ✓ | – | ✓ | ✓ | ✓ | ✓ | ✓ | **1**$^a$, 4, **8**$^a$, 14–16,18-21 |
| 14 | *MinimalΔL6L4* | ✓ | ✓ | ✓ | ✓ | – | – | – | – | – | ✓ | ✓ | 4, **12**, 14, 15, 18, 21 |

Rows 1–4, genes homologous to TropA–D produce only polyketides; Rows 5–8, full cluster expression, and omission of superfluous genes (*asl2, asL5* and *asR4*) produce **1**. Rows 9–14, expression of minimal cluster and omission of genes encoding catalytically active proteins. Bold numbers indicate meroterpenoids. $^a$ = trace amounts

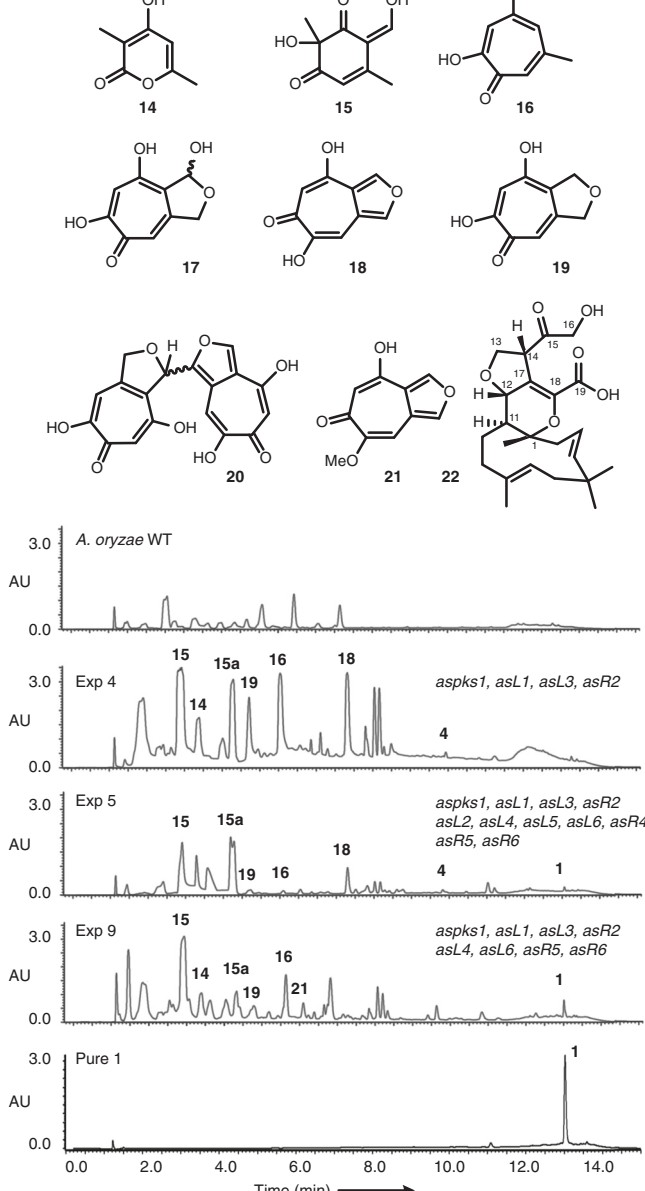

**Fig. 4** Compounds isolated from heterologous expression in *A. oryzae*. Diode array (DAD) data are shown. See Supplementary Figs. 50–53 for chromatograms for all other expression experiments

although the early tropolone biosynthetic steps are clearly still functional.

AsL6 and AsL4 appear to be able to partially complement each other's activities as xenovulene biosynthesis was severely reduced, but not abolished in each individual KOe experiment (Table 2, exp. 12 and 13). In these experiments, phenolic meroterpenoids **7** (ΔasL6) and **8** (ΔasL4) were observed, albeit in low titres. The same compounds were also observed in WT *S. schorii*. In a final deletion experiment, the minimal gene set lacking both *asL6* and *asL4* was expressed, resulting in production of **12** in good titre, but no other meroterpenoids were observed (Table 2, exp. 14).

**In vitro investigations**. The biosynthetic data gathered by Simpson and coworkers[3] strongly suggest that humulene **2** forms the terpene moiety of **1**. However, although humulene **2** is a common component of plant essential oils, only the sesquiterpene cyclase from *Zingiber zerumbet* (bitter ginger) has been reported

to produce humulene as a major product[26,27]. Humulene **2** is rarely reported as a metabolite of fungi, and is known only in *Fusarium fujikuroi* (Ffsc4)[28], *Colletotrichum acutatum* (CaTPS)[29] and *Stereum hirsutum*[30]. The class-I terpene cyclases Ffsc4 and CaTPS from these organisms were previously characterised by heterologous expression. Both are known to produce **2** alongside β-caryophyllene. Searches of the xenovulene BGC and the wider *S. schorii* genome using these sequences, however, were fruitless. No gene in the xenovulene biosynthetic cluster shows significant homology to any known terpene cyclase, and sesquiterpene cyclases discovered elsewhere on the *S. schorii* genome are not significantly expressed during xenovulene production. WT *A. oryzae* is not known to produce humulene and this led us to conclude that one of the two proteins of unknown function may possess this role.

To probe the biosynthetic steps further we attempted in vitro assay of the key steps. We synthesised intron-free *E. coli* optimised clones of *asR5*, *asR6*, *asL4* and *asL6* with *N*-terminal His$_6$ tags. The genes *asR5* and *asR6* were successfully expressed in *E. coli*, leading to the production of the expected proteins (48.5 kDa and 52.5 kDa, respectively, Supplementary Fig. 58) in soluble form. Attempts to obtain soluble preparations of AsL4 and AsL6 have not yet been successful.

Incubation of AsR6 with farnesylpyrophosphate **23** produced humulene **2** (GCMS analysis, Fig. 5, Supplementary Fig. 59). Production depends on the presence of Mg$^{2+}$ ions: either exchange for Ca$^{2+}$ or treatment with 20 mM EDTA abolishes activity. In vitro experiments to probe the role of AsR5 were inconclusive. Incubation of AsR5 with tropolone **5** and humulene itself did not produce any additional compounds and an alternative assay involving incubation of **5** with AsR5, AsR6 and FPP **23** also gave no new products.

## Discussion

These results show that the early steps in the biosynthesis of the xenovulenes are identical to those involved during the production of stipitatic acid **3**[4]. The BGC contains four genes with high homologies to the *tropA–D* genes from the *Talaromyces stipitatus* stipitatic acid **3** pathway, and heterologous expression experiments in *A. oryzae* showed that they catalyse the same set of reactions, via **15** and **5** to give the unstable lactol **17**, which is shunted in *A. oryzae* to **18**, **19**, **20** and **21**. Varying amounts of these metabolites obtained in different experiments reflect the unpredictable nature of these shunt pathways. Previous work in *A. oryzae* has shown that aldehydes in particular are rapidly shunted when the succeeding enzymatic steps are absent[31].

The key structural feature of xenovulene A **1** is the presence of the humulene sesquiterpene. Although humulene **2** has been reported as a fungal metabolite, searches of the *S. schorii* genome obtained here, using known humulene synthases, found no convincing matches. Our expression experiments, supported by in vitro assays, convincingly show that AsR6 is a terpene cyclase, which appears to be unrelated to other known terpene cyclases, at least in its primary structure. The well-understood magnesium ion-binding residues of class-I cyclases are not conserved in AsR6 (Supplementary Methods), and no other significant sequence homologies to known humulene or other sesquiterpene synthases are observed. In vitro assays, however, support a Mg$^{2+}$-dependent mechanism, but further analysis will have to await structural determination of AsR6 before further conclusions can be drawn. AsR6 shows high sequence homology to other fungal proteins of unknown function (Supplementary Methods and Supplementary Fig. 56) and thus may represent the first of a larger family of terpene cyclases which have not yet been explored. In one case, in the human pathogenic *Aspergillus thermomutatus* (*Neosartorya*

**Fig. 5** GCMS analysis of AsR6 activity (total ion current, arbitary units) of pentane extracts of in vitro assays. **a** Humulene (Sigma-Aldrich); **b** AsR6 + 10 mM HEPES, 5 mM Mg²⁺, 5 mM DTT and 150 μM FPP; **c** as **b**, lacking AsR6. See Supplementary Fig. 60 for MS data

*pseudofischeri*), this putative terpene cyclase gene is clustered with other homologues of the xenovulene cluster genes and may thus be involved in the biosynthesis of a to-date unknown tropolone meroterpenoid (Supplementary Fig. 57).

Our KOe experiments suggest that AsR5 is involved in the connection of the polyketide and tropolone moieties, as experiments lacking AsR5 were unable to make meroterpenoids, but lack of observable activity in vitro has precluded further conclusions for the time-being. The lack of activity may be related to the rapid conversion of putative lactol **17** to shunts. In vivo experiments in which AsR2 and AsR5 are present, form **12** (e.g. expt 14), consistent with this idea. Compound **12** is clearly closely related to the known tropolone meroterpenoids, including epolone B **24**, a μM inducer of human erythropoietin[32], and pycnidione[33] **25** a μM inhibitor of stromelysin (Fig. 6).

Intriguing biomimetic work reported by Baldwin and coworkers suggests that quino-methide tropolones similar to **26** (Fig. 7) react with humulene **2** to form meroterpenoids in a probable hetero-Diels–Alder (hDA) reaction[34]. The relative stereochemistry at the 1,11 positions of **12** and **1** itself is consistent with this idea. However, such reactions require very harsh conditions in vitro (e.g. 150 °C, 24 h) to form the required tropolone quinomethides **26** and one role of AsR5 may be to produce the required enone catalytically by elimination of water. In the absence of AsR5 this compound seems to be shunted to **18–21** (Fig. 7). AsR5 probably also catalyses the putative hDA reaction itself as racemic products would be expected from spontaneous reactions, whereas the xenovulenes (and other tropolone meroterpenoids such as **24** and **25**) are found as single stereoisomers. Several enzymatic Diels–Alder catalysts have been proven to be involved in other biosynthetic pathways, for example, during the biosynthesis of abyssomicin[35], but AsR5 shows no significant homology to any of these known enzymes. Tang and coworkers recently reported a putative intramolecular hetero-Diels Alderase that operates during the biosynthesis of leporin B[36]. However, the leporin B enzyme (LepI) requires SAM as a cofactor and bears no sequence similarity to AsR5 which has no recognisable cofactor-binding site, and that appears to catalyse an intermolecular reaction.

AsR5 also differs from other enzymes known to be involved in the biosynthesis of fungal meroterpenoids[37]. The previously described enzymes act by initially attaching a non-cyclised terpene to an aromatic polyketide (e.g. mycophenolic acid)[38], usually followed by oxidation and epoxide-mediated (e.g.

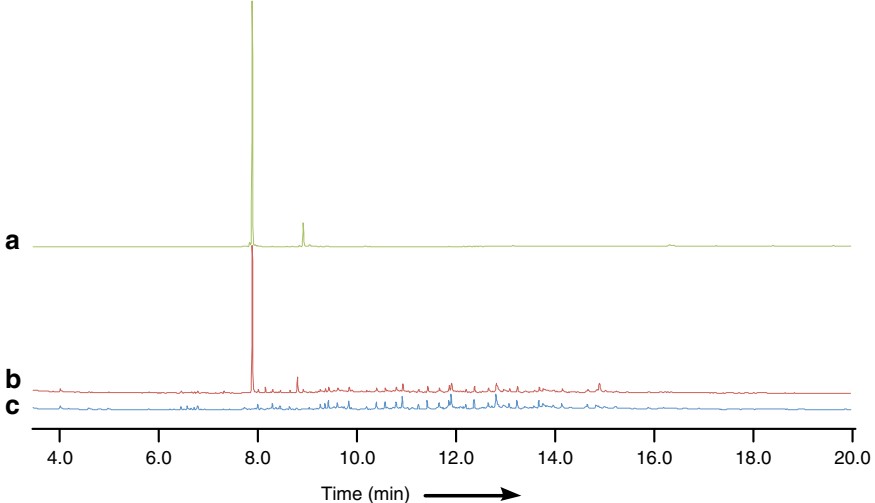

**Fig. 6** Comparison between isolated xenovulene precursor **12** and known tropolone meroterpenoids **24** and **25**. The figure shows the structural similarities between **12**, **24** and **25**

anditomin)[39] cyclisation, or acid-catalysed cyclisation (e.g. the macrophorins)[40]. During xenovulene biosynthesis, however, the terpene is cyclised before attachment to the polyketide. This observation is consistent with the fact that the post-PKS genes in the xenovulene BGC show no significant similarity to genes in other known fungal meroterpenoid BGCs.

Finally, our results show that AsL4 and AsL6 are previously unreported types of oxidative-ring-contracting enzymes. Homology analysis showed these proteins to be NAD/FAD-dependent enzymes that suggest their role in the required oxidative-ring contractions. KOe of either *asl4* or *asl6* reduced (but did not abolish) the biosynthesis of **1** in *A. oryzae*, where trace amounts could be observed. The previously observed **7** was detected in the *asL6* KOe experiment; while the isomeric **8** was observed in the KOe of *asL4* suggesting these as shunts or intermediates. Observation of the synthesis of **1** in each experiment shows that these proteins can partially complement each other. Disruption of both genes leads only to production of the initial meroterpenoid intermediate **12** and no further oxidations or ring contractions. Oxidative-ring contractions must also occur during the biosynthesis of terrein **6**, for example, but no significant homology between AsL4 and AsL6 and proteins encoded by the terrein **6** BGC[6] was observed.

Overall, our results support a biosynthetic pathway to xenovulene that involves the likely formation of the tropolone hemiacetal **17** by the already well-understood fungal tropolone

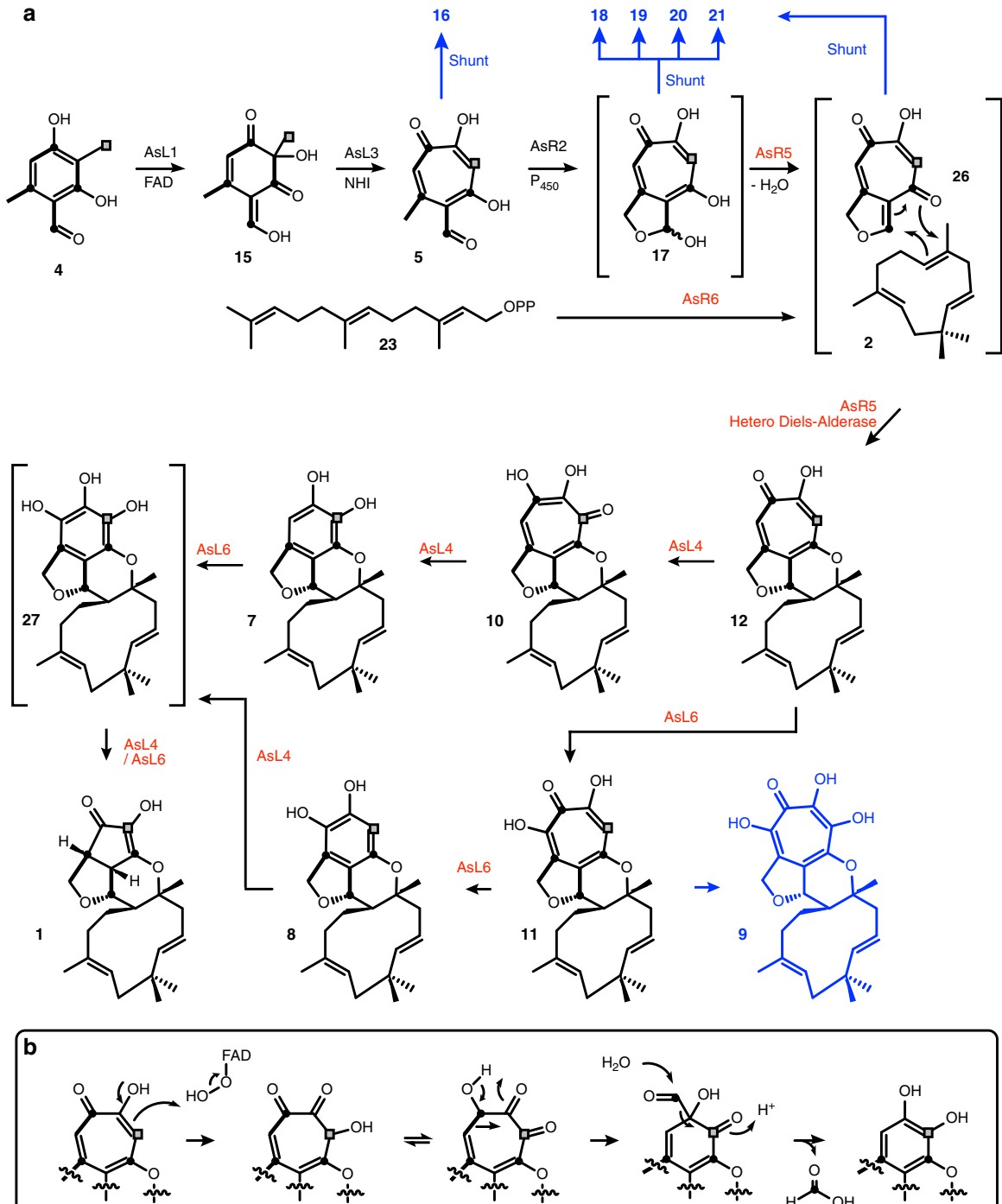

**Fig. 7** Overall pathway. **a** Proposed pathway to **1** and routes to shunt metabolites (blue). Compounds in brackets are not directly observed. Red steps are previously unknown. **b** Proposed mechanism for oxidative-ring contraction[3]. Labelling pattern (bold bond, intact acetate; grey square from S-adenosyl methionine [SAM]) as observed by Simpson and coworkers[3]

pathway (Fig. 7). In parallel, a previously unrecognised class of terpene cyclase, AsR6, produces humulene and a likely hetero Diels–Alder reaction catalysed by AsR5 would produce **12** as observed here in WT *S. schorii*. Oxidative-ring contractions catalysed by AsL4 and AsL6 then processively remove carbon atoms from the polyketide. Intermediates in this process could be **12**, **10** and **7** that are also observed in WT *S. schorii*, and **27** that has not yet been isolated. It is not clear if **8**, **9** and **11** are shunt metabolites or true intermediates and such studies will have to await successful in vitro experiments with AsL4 and AsL6. The

suggested pathway is in full agreement with the results of Simpson's earlier isotopic labelling studies.

Thus, the pathway reveals three highly unusual classes of biosynthetic enzymes in fungi, and reveals a previously unknown mode of meroterpenoid biosynthesis via a likely intermolecular hDA reaction. The isolation of **12** through partial pathway expression clearly links the biosynthesis of **1** to the important class of tropolone meroterpenoids, but further genetic analysis of the biosynthesis of these related compounds will have to await genome sequencing of the producing organisms.

## Methods

**Fermentation and extraction protocols**. *S. schorii*: Hundred-millilitre *S. schorii* cultures were centrifuged (9000 × *g*, 10 min), the supernatant was acidified to pH 2 with HCl (2 M) and extracted twice with ethyl acetate/hexane (1:1, 2 × 100 mL). The combined organic layers were dried over MgSO₄ and solvents were removed in vacuo. The resulting extract was dissolved in methanol or acetonitrile:water (9:1) to a concentration of 10 mg/mL, filtered through glass wool and analysed by LCMS.

*A. oryzae* NSAR1: Hundred-millilitre *A. oryzae* NSAR1 cultures were acidified to pH 2 with HCl (2 M) and incubated for 30 min (28 °C, 110 rpm). Cells were disrupted with a hand blender and removed by Büchner filtration. The resulting supernatant was extracted twice with ethyl acetate or ethyl acetate/hexane (1:1, 2 × 100 mL). The combined organic layers were dried over MgSO₄ and solvents were removed in vacuo. The extract was dissolved in methanol to a concentration of 10 mg/mL, filtered over glass wool and analysed by LCMS.

**Analytical LCMS and HRMS**. LCMS data were obtained using a Waters LCMS system comprising of a Waters 2767 autosampler, Waters 2545 pump system and a Phenomenex Kinetex column (2.6 μ, C₁₈, 100 Å 4.6 × 100 mm) equipped with a Phenomenex Security Guard precolumn (Luna C₅ 300 Å) eluted at 1 mL/min. Detection was performed by Waters 2998 diode array detector between 200 and 600 nm; Waters 2424 ELSD and Waters SQD-2 mass detector operating simultaneously in ES⁺ and ES⁻ modes between 100 m/z and 650 m/z. Solvents were **A**, HPLC-grade H₂O containing 0.05% formic acid; and **B**, HPLC-grade CH₃CN containing 0.045% formic acid. Gradients were as follows: Method 1 (optimised for non-polar compounds): 0 min, 10% **B**; 10 min, 90% **B**; 12 min, 90% **B**; 13 min, 10% **C** and 15 min, 10% **B**. Method 2 (optimised for polar compounds): 0 min, 10% **B**; 10 min, 40% **B**; 12 min, 90% **B**; 13 min, 10% **B** and 15 min, 10% **B**. HRMS was obtained using a UPLC system (Waters Acquity Ultraperformance, running the same method and column as above) connected to a Q-TOF Premier mass spectrometer.

**Preparative LCMS and NMR**. Compounds were purified using a Waters mass-directed autopurification system consisting of a Waters 2545 pump and Waters 2767 autosampler. The chromatography column was a Phenomenex Kinetex Axia column (5 μ, C₁₈, 100 Å, 21.2 × 250 mm) fitted with a Luna C₅ 300 Å Phenomenex Security Guard precolumn. The column was eluted at 20 mL/min at 22 °C. Solvents used were **A**, H₂O + 0.05% formic acid; and **B**, CH₃CN + 0.045% formic acid. All solvents were HPLC grade. The column outlet was split (100:1) and the minority flow was supplemented with HPLC-grade MeOH + 0.045% formic acid to 1000 μL/min and diverted for interrogation by diode array (Waters 2998) and evaporative light-scattering (Waters 2424) detectors. The flow was also analysed by mass spectrometry (Waters SQD-2 in ES⁺ and ES⁻ modes). Desired compounds were collected into glass test tubes. Combined fractions were evaporated in vacuo, weighed and dissolved directly in NMR solvent for analysis.

A Bruker Avance 500 instrument equipped with a cryo-cooled probe at 500 MHz (¹H) and 125 MHz (¹³C) was used for all NMR analysis. Standard parameters were used for the collection of 2D spectra (¹H, ¹H-correlation spectroscopy [COSY], heteronuclear single-quantum coherence [HSQC] and HMBC) in the indicated solvents. ¹H and ¹³C spectra are referenced relative to residual protonated solvents. All δ values are quoted in ppm and all *J* values in Hz.

**S. schorii KO procedure**. A well-grown *S. schorii* single colony was used to inoculate 100 mL (500-mL flask) of ASSM (10 g/L glucose, 15 g/L glycerol, 15 g/L polypeptone, 3 g/L NaCl, 5 g/L malt extract, 1 g/L Junlon PW110 and 1 g/L Tween80) liquid culture and incubated for 1.5 days (25 °C, 200 rpm). Cells were collected by filtration over sterile miracloth, washed with 0.8 M NaCl (50 mL) and suspended in a 10-mL filter sterilised with protoplasting solution (20 mg/mL lysing enzyme from *Trichoderma harzianum*, Sigma-Aldrich, 10 mg/mL driselase from Basidiomycetes sp., Sigma-Aldrich, 0.8 M NaCl and 10 mM K₂HPO₄/KH₂PO₄ at pH 6.0). The suspension was incubated at 30 °C and 150 rpm up to 1.5 h. Protoplasts were released by pipetting, collected by centrifugation (2000 × *g*, 5 min) and washed twice with 20 mL of 0.8 M NaCl (2000 × *g*, 5 min). Protoplasts were suspended in solution I (10 mM CaCl₂, 0.8 M NaCl and 50 mM Tris-HCl at pH 7.5). DNA (≥1 μg, in 10 μL of ddH₂O) was mixed with 100 μL of protoplasts (10⁷) and incubated on ice for 5 min. An aliquot of 1 mL of fungal transformation solution II (50 mM CaCl₂, 0.8 M NaCl, 50 mM Tris-HCl at pH 7.5 and 60% (w/v) poly-ethyleneglycol [PEG] 3350) was added and the mixture was incubated at room temperature for 20 min. Thirty millilitres of molten CD + S 0.8% agar (~37 °C) with 100 μg/mL hygromycin B was added and the mixture was poured over two plates. The plates were incubated at 25 °C until resistant colonies could be observed, and these were transferred to secondary plates of CD + S 1.5% agar containing 100 μg/mL hygromycin B. Vigorously growing colonies were transferred onto the third plate (CD + S 1.5% agar, 100 μg/mL hygromycin B) after 7 d and streaked for single colonies. All used CD + S media were prepared with deionised water, as no colonies could be recovered with Millipore water.

**A. oryzae transformation procedure**. One-millilitre spore suspension from a fresh *A. oryzae* NSAR1 DPY plate was used to inoculate 100 mL (500-mL flask) of GN liquid culture and incubated for 12 h (28 °C, 180 rpm). Cells were collected by filtration over sterile miracloth, washed with 0.8 M NaCl (50 mL) and suspended in 10 mL of filter-sterilised *A. oryzae* NSAR1 protoplasting solution (20 mg/mL lysing enzyme from *Trichoderma harzianum*, Sigma-Aldrich, 0.8 M NaCl, 10 mM). The suspension was incubated at 2 h (30 °C, 150 rpm). Protoplasts were released by pipetting, collected by centrifugation (2000 × *g*, 5 min) and directly suspended in the required amount of fungal transformation solution I (10 mM CaCl₂, 0.8 M NaCl and 50 mM Tris-HCl at pH 7.5). DNA (≥1 μg, in 10 μL of ddH₂O) was mixed with 100-μL protoplasts and incubated on ice for 5 min. One millilitre of fungal trans-formation solution II (50 mM CaCl₂, 0.8 M NaCl and 50 mM Tris-HCl at pH 7.5, 60% (w/v) PEG3350) was added and the mixture was incubated at room temperature for 20 min. Fourteen millilitres of molten CZD + S soft agar (~37 °C; 3.5% CD, 1 M sorbitol, 0.1% ammonium sulphate and 0.8% agar) was added and the mixture was poured over two plates containing CZD + S agar (3.5% CD, 1 M sorbitol, 0.1% ammonium sulphate and 1.5% agar). Plates were incubated at 28 °C until colonies could be observed, and these were transferred to secondary plates of CZD agar (3.5% CD, 0.1% ammonium sulphate and 0.8% agar). Vigorously growing colonies were transferred onto the third plate (CZD 1.5% agar) and streak purified.

**In vitro assay and analysis of AsR6**. Enzymatic activity of AsR6 was determined by incubation of AsR6 (5 μM) with farnesylpyrophosphate (150 μM), magnesium chloride (5 mM) or calcium chloride (5 mM), dithiothreitol (5 mM) and 10 mM HEPES buffer (pH 7.5) for 30 min at 28 °C. Optionally ethylenediaminetetraacetic acid (EDTA) was added to a final concentration of 20 mM. Enzymatic reactions were extracted with 300 μL of n-pentane and extracts were analysed directly by GC–MS. For GC–MS analysis, a HP 6890 gas chromatograph connected to a 5973 mass detector (Agilent) was used. The GC system was equipped with an OPTIMA 5 MS capillary column (30 m, 0.32 mm i.d., 0.25-μm film). Instrumental parameters were **1**, He at 1.5 mL/min; **2**, injection volume, 5 μL; **3**, transfer line, 280 °C and **4**, electron energy, 70 eV. GC programme: 1 min at 50 °C prior to increasing the temperature at 20 °C/min to 300 °C. The GC was operated in splitless mode.

**Data availability**. All NMR data, details of cloning procedures, detailed LCMS chromatograms and details of bioinformatic procedures and results are contained in the Supplementary Information. The xenovulene BGC is deposited at GenBank with accession number MG736817. All other data are available from the authors upon reasonable request.

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

## Acknowledgements

We thank DFG (INST 187/621) for LCMS equipment. R.S. and C.S. are funded by The Leibniz Universität Hannover. The bioinformatics support of the BMBF-funded project Bielefeld-Gießen Center for Microbial Bioinformatics-BiGi (grant number 031A533) within the German Network for Bioinformatics Infrastructure (de.NBI) is gratefully acknowledged. We thank Miriam Streeck and Katja Körner for technical assistance. We thank Clara Oberhauser and Professor Andreas Kirschning for the gift of FPP and the Leibniz University of Hannover and Institute of Organic Chemistry for financial and technical support.

## Author contributions

R.J.C. designed the study and wrote the manuscript. R.S. performed all experimental procedures and collected all data, except those performed by C.S. C.S. expressed *E. coli* optimised genes in *E. coli*, purified proteins and performed the in vitro assays. R.S. and C.S. prepared the experimental and supplementary material. D.W. and J.K. performed the genome and transcriptome sequencing and annotation.

## Additional information

**Competing interests:** The authors declare no competing interests.

