## [Peer Review File · Nature Communications]

Reviewers' comments:

Reviewer #1 (Remarks to the Author):

The manuscript by Schor et al. describes the identification and functional analysis of the xenovulene A biosynthetic gene cluster from *A. strictum* (*Sarocladium schorii*). The authors identified seven xenovulene A related compounds, compound 7-13, and their chemical structures were elucidated by LCMS, UV, and NMR analysis. Genome sequencing revealed that *S. schorii* has 39 putative biosynthetic gene clusters. Among them, the 49 kb DNA fragment was identified to be a xenovulene biosynthetic gene cluster, because it carries genes encoding homologue enzymes (Aspks1, AsL1, AsL3, and AsR2) for tropolon biosynthesis. In fact, the authors demonstrated that the knockout mutant of the aspks1 gene did not produce xenovulene and its related compounds. RNAseq analysis also revealed that the genes from asL7 to asR7 are active under the xenovulene producing condition. Heterologous host (*A. oryzae*) harbouring the gene cluster produced xenovulene and biosynthetic intermediates, demonstrating that the gene cluster indeed encodes xenovulene biosynthesis. In addition, the minimal set of genes responsible for the biosynthesis of xenovulene was determined by the heterologous expression experiments as follows: aspks1; asL1 encoding an FAD dependent hydroxylase; asL3 encoding a non-heme iron dioxygenase; asR2 encoding a cytochrome P450 monooxygenase; asR5 and asR6 encoding proteins of unknown function; and asL6 and asL4 encoding putative NAD/FAD dependent oxidoreductases. The recombinant enzyme of AsR6 (rAsR6), a new type of terpene cyclase, produced humulene using FPP as a substrate. Although the authors were unable to detect the enzymatic activity of rAsR5, it is strongly suggested to be involved in the connection of the polyketide and tropolone moieties. The findings from the heterologous expression experiments also supports the suggested function (catalysing Diels-Alder reaction). Thus, this manuscript demonstrated the biosynthetic pathway for xenovulene for the first time and unique functions of the biosynthetic enzymes. Overall, this manuscript would be a nice contribution for publication in Nature Communications. However, it was hard to understand the figures and tables due to poor figure legends. In addition, you need to link the figures/tables of Supplemental materials to the main text.

Other minor clarifications/edits are suggested to improve the manuscript as follows:

Page 2, line 37 and 42

Scheme 1A? The authors need to check whether it is correct. It must be Figure 1.

Page 3, line 56, 64, 68, and 73

The abbreviations, CABI, HRMS, HMBC, and ESI should be defined here.

Reviewer #2 (Remarks to the Author):

A great amount of work characterizing products and shunt products from heterologous expression of various combinations of genes from the xenovulene BGC isolated from *A. strictum* is described in this manuscript. Their structure proofs appear meticulous and accurate (and thoroughly documented in the SI). The discussion does a good job applying chemical logic and precedent to assign what is main pathway and what is not. Based (I would guess from the presentation) on chemical precedent from Baldwin, a hetero Diels-Alder reaction is proposed to be catalyzed by AsR5 between hypothesized intermediate 26 and humulene. What is to eliminate ionization of 17, however, to an oxonium, attack by humulene to give a 3° cation, which traps the tropolone alcohol to the product (I believe a C=C is missing in 12 in Sch. 1)? Stepwise ionic mechanisms by enzymes are better precedented. An interesting ring contraction is proposed to occur between 10 and 7. No mechanism is suggested for how this might take place. Finally, AsR6, which does not

align with normal prenyl cyclases converts FPP (requirement for Mg⁺⁺ was shown). All discussion of how this enzyme might function is deferred to an eventual crystal structure.

All of this is good as far as it goes, but the lack of follow through is frustrating for the three highlighted enzyme transformations. Is this better as a nice *Angewandte* summary introduction to a set of interesting reactions to be followed up with more detailed papers? Or could the authors bring a deeper discussion and consideration of alternative mechanisms to these processes that I more associate with the *Nature* family or journals? This is a matter of opinion about "appropriateness" where the Editors are better judges than I, but I am feeling short-changed.

Reviewer #3 (Remarks to the Author):

This manuscript describes the identification and partial functional characterization of the biosynthetic pathway for the production of the meroterpenoid xenovulene by *Acremonium strictum*. Previously, the corresponding author and others have identified enzymes involved in the formation and rearrangement of a polyketide moiety that is proposed to be transformed into the polyketide moiety found in xenovulene. In this study, genome sequencing of the xenovulene producing fungus lead to the identification of a putative biosynthetic gene cluster containing homologs of these known enzymes, which was confirmed by deleting the known, scaffold forming polyketide synthase. Combinatorial expression of cluster genes in *Aspergillus* and structural identification of produced metabolites was used to deduce the functions of individual genes. Two genes are proposed to encode a novel type of sesquiterpene cyclase and Hetero-Diels Alderase. Function of the two enzymes have been proposed based on gene expression studies in *Aspergillus*, and partially using a recombinant, purified protein for the cyclase.

Considerable research efforts went into this work, yielding potentially important new results on a novel cyclase. However, the manuscript is very poorly organized and written, making it difficult to follow the bewildering number of compounds, gene combinations and schemes throughout the manuscript. Figure and Scheme legends are sparsely explained and in some instances, do not even match how they are referred to in the text (e.g. Figure 1 and Schem1A-C, does not make sense). Panels are not numbered and explained. Table 2 has strange symbols (also in the text) and in the legend, there is a reference to "inactive genes" – which does not make sense.

As such, I strongly suggest that the authors completely rewrite and reorganize the manuscript. The introduction seems to be merged together from another draft (with Scheme 1?). Likewise, the first part describing in detail the structural identification of compounds seems to be more appropriate for a more chemistry oriented journal. Some of these details can be relegated into the Supporting Information (which should be combined into one document, and not referred to as "ESI" – mistook this first as Electrospray ionization).). Organize the manuscript into subsection such as e.g. Meroterpenoid production by *A. strictum*, Genome sequence and Cluster annotation, Functional identification of biosynthetic genes, In vitro characterization of...

I also strongly suggest combining, if possible, all structures into one master scheme (including the shunt pathway products) – and not hide Scheme 1 in the "Discussion". It would it make much easier to follow the experiments and especially, the heterologous expression studies.

What is the rationale for renaming *A. strictum*? This does not appear to be warranted.

Figure 1: I do not quite understand why the labeling patterns are relevant here. They have been described elsewhere. Provide panel descriptions and labels.

The description of the cluster characterization is confusing. Why not describe it in the order as it has probably been done experimentally – one omission of on individual gene at the time and finally, multiple genes, including those that appear to have no function in meroterpenoid biosynthesis. Several constructs, however, seems to be missing – co-expression of the minimal set

of four genes with asR5 or asR6, and co-expression of the six functional genes with asR5 or asR6.

Metabolite profiles in Figures 1 and 3 could be combined in one Figure and each chromatogram labeled with the corresponding strain (wildtype, *Aspergillus* and gene combinations etc. – and not just the experiment)

Finally, my major concern relates to the characterization of the sesquiterpene cyclase activity. First, looking at the SDS-PAGE of the purified protein, it does not appear to be very pure. From the GC-MS trace, the enzyme seems have very poor activity. Compared to the control reaction, the humulene peak is not much higher than the background peaks. It is surprising that the reaction does not give a cleaner trace – what is the Y-axis? Abundance? Provide mass spectra to confirm humulene structure.

Assuming that the enzyme has been expressed in *E. coli* and it acts on FPP, the *E. coli* cultures should also produce humulene as has been shown for other sesquiterpene. Please show a production profile for *E. coli* as well. For comparison and proof that this is a bona fide sesquiterpene synthase, the authors must perform kinetic measurements for this enzyme (similar experiments were performed for other novel cyclase activities, e.g. the HAD-like cyclase enzyme).

Apart from the *in vivo* expression studies, there is unfortunately little proof that asR6 encodes a Diels-Alderase. The authors should be careful in assigning this function to this gene. There may well be other enzymes involved. This statement should be toned down until they have clear biochemical evidence.

Reviewers' comments:

Reviewer #1 (Remarks to the Author):

The manuscript by Schor et al. describes the identification and functional analysis of the xenovulene A biosynthetic gene cluster from *A. strictum* (*Sarocladium schorii*). The authors identified seven xenovulene A related compounds, compound 7-13, and their chemical structures were elucidated by LCMS, UV, and NMR analysis. Genome sequencing revealed that *S. schorii* has 39 putative biosynthetic gene clusters. Among them, the 49 kb DNA fragment was identified to be a xenovulene biosynthetic gene cluster, because it carries genes encoding homologue enzymes (*Aspks1*, *AsL1*, *AsL3*, and *AsR2*) for tropolon biosynthesis. In fact, the authors demonstrated that the knockout mutant of the *aspks1* gene did not produce xenovulene and its related compounds. RNAseq analysis also revealed that the genes from *asL7* to *asR7* are active under the xenovulene producing condition. Heterologous host (*A. oryzae*) harbouring the gene cluster produced xenovulene and biosynthetic intermediates, demonstrating that the gene cluster indeed encodes xenovulene biosynthesis. In addition, the minimal set of genes responsible for the biosynthesis of xenovulene was determined by the heterologous expression experiments as follows: *aspks1*; *asL1* encoding an FAD dependent hydroxylase; *asL3* encoding a non-heme iron dioxygenase; *asR2* encoding a cytochrome P450 monooxygenase; *asR5* and *asR6* encoding proteins of unknown function; and *asL6* and *asL4* encoding putative NAD/FAD dependent oxidoreductases. The recombinant enzyme of *AsR6* (*rAsR6*), a new type of terpene cyclase, produced humulene using FPP as a substrate. Although the authors were unable to detect the enzymatic activity of *AsR5*, it is strongly suggested to be involved in the connection of the polyketide and tropolone moieties. The findings from the heterologous expression experiments also supports the suggested function (catalysing Diels-Alder reaction).

Thus, this manuscript demonstrated the biosynthetic pathway for xenovulene for the first time and unique functions of the biosynthetic enzymes. Overall, this manuscript would be a nice contribution for publication in Nature Communications.

However, it was hard to understand the figures and tables due to poor figure legends.

The figures have been rationalised: Previous Figure 1 has been split in two; Scheme 1 is now renamed Figure 6 for simplicity; Figures have been given clearer panel labels (*A*, *B*, *C* etc) and legends have been expanded to include more detail.

In addition, you need to link the figures/tables of Supplemental materials to the main text.

This has been done.

Page 2, line 37 and 42

Scheme 1A? The authors need to check whether it is correct. It must be Figure 1.

This has been changed

Page 3, line 56, 64, 68, and 73

The abbreviations, CABI, HRMS, HMBC, and ESI should be defined here.

This has been done. ESI has been changed to 'supplementary information' throughout and links are now included to supplementary information tables, figures and section numbers.

Reviewer #2 (Remarks to the Author):

A great amount of work characterizing products and shunt products from heterologous expression of various combinations of genes from the xenovulene BGC isolated from *A. strictum* is described in this manuscript. Their structure proofs appear meticulous and accurate (and thoroughly documented in the SI). The discussion does a good job applying chemical logic and precedent to assign what is main pathway and what is not. Based (I would guess from the presentation) on chemical precedent from Baldwin, a hetero Diels-Alder reaction if proposed to be catalyzed by AsR5 between hypothesized intermediate 26 and humulene. **What is to eliminate ionization of 17, however, to an oxonium, attack by humulene to give a 3° cation, which traps the tropolone alcohol to the product (I believe a C=C is missing in 12 in Sch. 1)? Stepwise ionic mechanisms by enzymes are better precedented.**

Stepwise or concerted? The eternal question of biological Diels Alder reactions! Either mechanism is conceptually satisfying, but obtaining proof (or even good evidence) either way is notoriously difficult. Here we are unable to access the likely true substrate, the tropolone quino-methide 26, because when AsR5 is absent from the expression, 26 is shunted to 18-21. Likewise 17 is not isolatable, and 5 is not a substrate, so probing this question is not yet possible. In Baldwin's earlier *in vitro* work (Baldwin, J., Mayweg, A., Neumann, K. & Pritchard, G. Studies toward the biomimetic synthesis of tropolone natural products via a hetero Diels-Alder reaction. *Org. Lett.*, 1, 1933-1935 (1999).) the quinomethide was generated by high temperature in the presence of 2 and absence of water, but when formed it rapidly reacts with 2. Note that 'ionisation of 17' gives a product which is just a resonance form of protonated 26.... The role of the enzyme may well be to generate such intermediates, but significant further work will be required to probe this and I did not feel that speculation here would be constructive.

An interesting ring contraction is proposed to occur between 10 and 7. **No mechanism is suggested for how this might take place.**

The suggested mechanism is shown in Figure 6B (previously scheme 1)

Finally, AsR6, which does not align with normal prenyl cyclases converts FPP (requirement for Mg⁺⁺ was shown). All discussion of how this enzyme might function is deferred to an eventual crystal structure.

All of this is good as far as it goes, but the lack of follow through is frustrating for the three highlighted enzyme transformations. Is this better as a nice *Angewandte* summary introduction to a set of interesting reactions to be followed up with more detailed papers? **Or could the authors bring a deeper discussion and consideration of alternative mechanisms to these processes that I more associate with the *Nature* family or journals?**

I could indeed include more discussion - but this would be almost entirely speculative at this stage. The observations reported are supported by sound experimental data, but in the absence of soluble or active protein or specific substrates, no further hard evidence is yet available - this will form the focus of significant future laboratory efforts, of course. The purpose of publication in *Nature Communications* is to reach as wide an audience as quickly as possible for these new and unprecedented observations. We, and others, will no doubt build on the observations in future.

This is a matter of opinion about "appropriateness" where the Editors are better judges than I, but I am feeling short-changed.

Reviewer #3 (Remarks to the Author):

This manuscript describes the identification and partial functional characterization of the biosynthetic pathway for the production of the meroterpenoid xenovulene by *Acremonium strictum*. Previously, the corresponding author and others have identified enzymes involved in the formation and rearrangement of a polyketide moiety that is proposed to be transformed into the polyketide moiety found in xenovulene. In this study, genome sequencing of the xenovulene producing fungus lead to the identification of a putative biosynthetic gene cluster containing homologs of these known enzymes, which was confirmed by deleting the known, scaffold forming polyketide synthase. Combinatorial expression of cluster genes in *Aspergillus* and structural identification of produced metabolites was used to deduce the functions of individual genes. Two genes are proposed to encode a novel type of sesquiterpene cyclase and Hetero-Diels Alderase. Function of the two enzymes have been proposed based on gene expression studies in *Aspergillus*, and partially using a recombinant, purified protein for the cyclase.

Considerable research efforts went into this work, yielding potentially important new results on a novel cyclase. **However, the manuscript is very poorly organized and written, making it difficult to follow the bewildering number of compounds, gene combinations and schemes throughout the manuscript. Figure and Scheme legends are sparsely explained and in some instances, do not even match how they are referred to in the text (e.g. Figure 1 and Schem1A-C, does not make sense). Panels are not numbered and explained. Table 2 has strange symbols (also in the text) and in the legend, there is a reference to "inactive genes" - which does not make sense.**

The figures have been reworked to both clarify and include more detail. The legends are now clearer and no distinction is made between Figures and schemes (all are now figures for clarity). The figure panels are now more clearly labelled and linked to the text. There are no strange symbols in my version of Table 2 and I apologise if the journal pdf-conversion process has introduced these. Much clearer links to the supplementary information are now provided and in addition to the section subheadings now provided I believe the MS is much more digestible. The phrase *inactive genes* has been clarified to *superfluous genes*.

As such, I strongly suggest that the authors completely rewrite and reorganize the manuscript. The introduction seems to be merged together from another draft (with Scheme 1?). Likewise, the first part describing in detail the structural identification of compounds seems to be more appropriate for a more chemistry oriented journal.

The clear structural identification is crucial for the later parts of the manuscript where many of the compounds appear as products of various expression experiments. I believe it is important to first prove

the meticulous characterisation of these compounds (appreciated by reviewer 2 and accepted by reviewer 1) before the impact of the work summarised in Table 2 is discussed.

Some of these details can be relegated into the Supporting Information (which should be combined into one document, and not referred to as “ESI” – mistook this first as Electrospray ionization).

The supplementary information is a single document.

Organize the manuscript into subsection such as e.g. Meroterpenoid production by *A. strictum*, Genome sequence and Cluster annotation, Functional identification of biosynthetic genes, In vitro characterization of...

This has been done.

I also strongly suggest combining, if possible, all structures into one master scheme (including the shunt pathway products) – and not hide Scheme 1 in the “Discussion”. It would make much easier to follow the experiments and especially, the heterologous expression studies.

Scheme 1 is not 'hidden' and it does already synthesise all the structures and relationships in the MS.

What is the rationale for renaming *A. strictum*? This does not appear to be warranted.

Fungal isolate IMI 501407 was evidently mis-identified originally as *Acremonium strictum*. Microscopically it may appear to be very similar to other strains of *A. strictum*, but by ITS sequence, or more thoroughly by ANI analysis over the entire genome, it is clearly different and should be regarded as a separate species with a separate name.

Figure 1: I do not quite understand why the labeling patterns are relevant here. They have been described elsewhere. Provide panel descriptions and labels.

This has been done. However labelling pattern remains in Scheme 6B because it is important to show that the observed pathway is consistent with Simpson's original labelling results.

The description of the cluster characterization is confusing. Why not describe it in the order as it has probably been done experimentally – one omission of an individual gene at the time and finally, multiple genes, including those that appear to have no function in meroterpenoid biosynthesis.

The cluster characterisation is summarised in Table 2. Indeed this does progress logically: experiments 1-4 show the steps to tropolones; then genes from the entire genomic region were expressed (experiment 5) and non-participating genes (asL2, asL5 and asR4) are identified (expts 6,7, and 8). The function of the minimal cluster (exp 9) then shows that (asL2, asL5 and asR4) cannot complement one another. Then individual KO's of the genes of unknown function (asR5, asR6, asL6, asL4) are achieved to probe the individual steps (expts 10-13). Finally asL4 and asL6 are shown to partially complement one another (exp 14).

Several constructs, however, seem to be missing – co-expression of the minimal set of four genes with asR5 or asR6, and co-expression of the six functional genes with asR5 or asR6.

The minimal gene set consists of 8 genes - I assume the reviewer refers to the 4 genes which produce 17 (shunted to 18-21) above? This is experiment 14. Coexpression of the 6 functional genes with R5 and R6 is experiment 9.

Metabolite profiles in Figures 1 and 3 could be combined in one Figure and each chromatogram labeled with the corresponding strain (wildtype, *Aspergillus* and gene combinations etc. – and not just the experiment)

The profiles cannot be combined as different HPLC programmes were used (in order to separate the more non-polar components in Fig 1). I have added the gene combinations to the figures.

Finally, my major concern relates to the characterization of the sesquiterpene cyclase activity. First, looking at the SDS-PAGE of the purified protein, it does not appear to be very pure.

This has now been significantly improved with added purification steps detailed in the supplementary information (section S8).

From the GC-MS trace, the enzyme seems have very poor activity. Compared to the control reaction, the humulene peak is not much higher than the background peaks. It is surprising that the reaction does not give a cleaner trace – what is the Y-axis? Abundance? Provide mass spectra to confirm humulene structure.

The original figure was not clear - the experimental humulene peak was overlaid by the control peak and thus looked smaller than it was. The data is now represented more clearly.

Assuming that the enzyme has been expressed in *E. coli* and it acts on FPP, the *E. coli* cultures should also produce humulene as has been shown for other sesquiterpene. Please show a production profile for *E. coli* as well.

This has not been done as it adds nothing to the conclusion that purified AsR6 produces humulene from FPP.

For comparison and proof that this is a bona fide sesquiterpene synthase, the authors must perform kinetic measurements for this enzyme (similar experiments were performed for other novel cyclase activities, e.g. the HAD-like cyclase enzyme).

We have measured kinetic data for the sesquiterpene cyclase AsR6 between 0 and 200 μM FPP. The enzyme appears to suffer from extreme substrate inhibition (see below) and thus Michaelis-Menten kinetic parameters cannot be obtained. I would also argue that the collection of kinetic parameters in themselves does not prove that the enzyme is a terpene cyclase: the observation of conversion of FPP to humulene *in vitro* proves this assertion.

Apart from the *in vivo* expression studies, there is unfortunately little proof that asR6 encodes a Diels-Alderase. The authors should be careful in assigning this function to this gene. There may well be other enzymes involved. This statement should be toned down until they have clear biochemical evidence.

The language is moderated in the MS and the term is preceded by *likely* and *probable*.

REVIEWERS' COMMENTS:

Reviewer #1 (Remarks to the Author):

The revised manuscript by Schor et al has been adjusted based on the reviewer's comments. The authors have adequately answered all the questions raised by the reviewers and have improved their manuscript, which is now acceptable for publication by Nature Communications. But, this reviewer found additional minor points as follows.

Page 8, line 5; (exp. 1, see ESI for chromatograms) should be (exp. 1 in Table 1). You need to add the information about a supplemental figure for this.

Reviewer #2 (Remarks to the Author):

Reviewer #3 had it right. The figures and captions made a complex, multi-level investigation difficult to understand. Remarkably little has been done to the text, but reworking the figures has greatly improved presentation and reader comprehension. I am agreeable to publication as it currently stands.

Some wee grammatical suggestions:

p. 3, line 10 from bottom: "A series...was..."

p. 6, line 12 from bottom: "A total...was..."

p. 11, line 11 from bottom: "proteins that are involved..."

p. 15, lines 11 and 13 from bottom: again which to that

There are more. Cox or an Editor should have a go at the text and SI.

Reviewer #3 (Remarks to the Author):

The revised manuscript is now much easier to follow. Major concerns have been addressed.

RE NCOMMS-17-32913B

REVIEWERS' COMMENTS:

Reviewer #1 (Remarks to the Author):

The revised manuscript by Schor et al has been adjusted based on the reviewer's comments. The authors have adequately answered all the questions raised by the reviewers and have improved their manuscript, which is now acceptable for publication by Nature Communications. But, this reviewer found additional minor points as follows.

Page 8, line 5; (exp. 1, see ESI for chromatograms) should be (exp. 1 in Table 1). You need to add the information about a supplemental figure for this.

This has been done;

Reviewer #2 (Remarks to the Author):

Reviewer #3 had it right. The figures and captions made a complex, multi-level investigation difficult to understand. Remarkably little has been done to the text, but reworking the figures has greatly improved presentation and reader comprehension. I am agreeable to publication as it currently stands.

Some wee grammatical suggestions:

- p. 3, line 10 from bottom: "A series...was..."
- p. 6, line 12 from bottom: "A total...was..."
- p. 11, line 11 from bottom: "proteins that are involved..."
- p. 15, lines 11 and 13 from bottom: again which to that

There are more. Cox or an Editor should have a go at the text and SI.

These have been done - I apologise if the subtleties of "which vs that" did not make it into my O-level English grammar curriculum - experienced at a "Bog-Standard Comp"!

Reviewer #3 (Remarks to the Author):

The revised manuscript is now much easier to follow. Major concerns have been addressed.